# Potentially Asymptomatic Infection of Monkeypox Virus: A Systematic Review and Meta-Analysis

**DOI:** 10.3390/vaccines10122083

**Published:** 2022-12-06

**Authors:** Prakasini Satapathy, Parimala Mohanty, Subhanwita Manna, Muhammad A. Shamim, Priyanka Prasad Rao, Arun Kumar Aggarwal, Jagdish Khubchandani, Aroop Mohanty, Behdin Nowrouzi-Kia, Vijay Kumar Chattu, Bijaya Kumar Padhi, Alfonso J. Rodriguez-Morales, Ranjit Sah

**Affiliations:** 1Department of Virology, Postgraduate Institute of Medical Education and Research, Chandigarh 160012, India; 2Department of Community Medicine, Institute of Medical Sciences, SUM Hospital, Siksha ‘O’ Anusandhan Deemed to be University, Bhubaneswar 751003, India; 3Indian Institute of Public Health, Delhi 122002, India; 4Department of Pharmacology, All India Institute of Medical Sciences, Jodhpur 342005, India; 5Indian Institute of Public Health, Gandhinagar 382042, India; 6Department of Community Medicine, School of Public Health, Postgraduate Institute of Medical Education and Research, Chandigarh 160012, India; 7Department of Public Health, New Mexico State University, Las Cruces, NM 88003, USA; 8All India Institute of Medical Sciences, Gorakhpur 273008, India; 9ReSTORE Lab, Department of Occupational Science & Occupational Therapy, Temerty Faculty of Medicine, University of Toronto, Toronto, ON M5G 1V7, Canada; 10Center for Transdisciplinary Research, Saveetha Dental College, Saveetha Institute of Medical and Technological Sciences, Saveetha University, Chennai 602105, India; 11Department of Community Medicine, Faculty of Medicine, Datta Meghe Institute of Medical Sciences, Wardha 442107, India; 12Grupo de Investigación Biomedicina, Faculty of Medicine, Fundacion Universitaria Autonoma de las Americas—Institucion Universitaria Vision de las Americas, Pereira 660003, Risaralda, Colombia; 13Master of Clinical Epidemiology and Biostatistics, Universidad Cientifica del Sur, Lima 15024, Peru; 14Gilbert and Rose-Marie Chagoury School of Medicine, Lebanese American University, Beirut P.O. Box 36, Lebanon; 15Tribhuvan University Teaching Hospital, Institute of Medicine, Kathmandu 44600, Nepal; 16Department of Global Health and Clinical Research, Harvard Medical School, Boston, MA 02115, USA; 17Dr. D.Y. Patil Medical College, Hospital and Research Centre, Dr. D.Y. Patil Vidyapeeth, Pune 411018, India

**Keywords:** monkeypox, asymptomatic, infection, transmission, meta-analysis

## Abstract

**Background:** Monkeypox is a global public health concern, given the recent outbreaks in non-endemic countries where little scientific evidence exists on the disease. Specifically, there is a lack of data on asymptomatic monkeypox virus infection. This study aims to evaluate the overall prevalence of asymptomatic monkeypox virus infection. **Methods:** In this systematic review and meta-analysis, we performed an extensive literature search in PubMed, Scopus, Web of Science, ProQuest, EMBASE, EBSCOHost, Cochrane, and preprint servers (medRxiv, arXiv, bioRxiv, BioRN, ChiRxiv, ChiRN, and SSRN) and assessed all published articles till September 2022. Primary studies reporting monkeypox infections among asymptomatic participants were included after quality assessment. The characteristics of the study and information on the number of cases and symptomatic status were extracted from the included studies. The heterogeneity between studies was assessed using the I^2^ statistic. Publication bias was analyzed using funnel plots and Egger regression tests. The primary outcome was the pooled prevalence of asymptomatic infections within the examined population. **Results:** A total of 16 studies were included for qualitative synthesis, while five studies, including 645 individuals, were included for quantitative synthesis. There was substantial heterogeneity between studies (I^2^ = 94.86%; *p* < 0.01), with a pooled percentage of asymptomatic infections in the studied population of 10.2% (95%CI, 2.5–17.9%). **Conclusion:** This meta-analysis suggests that many people infected with the monkeypox virus are asymptomatic and difficult to detect. Therefore, prompt detection of these cases of monkeypox virus and appropriate subsequent management is of utmost importance to global public health.

## 1. Background

There has been an unprecedented spread of the monkeypox virus (MPXV) worldwide in 2022. Since the eradication of Smallpox in 1980, which resulted in the worldwide cessation of smallpox immunization, MPXV has resurfaced as an Orthopoxvirus of public health concern beyond endemic regions in Africa [1,2]. Although this zoonotic agent was found in laboratory monkeys in Denmark in 1958, it has been causing human disease, mainly in Africa. It first appeared in 1970 and has remained endemic for decades in parts of West and Central Africa [1,2,3]. The first human case was identified in the year 1970; there have been sporadic cases of the disease reported in the Central part of the African Republic, Congo (Kinshasa), Cameroon, Liberia, Sudan, and Nigeria [3,4,5,6]. Unlike the cases reported earlier, the current outbreak of 2022 exhibits unusual symptoms and clinical patterns in the endemic locations [3,4,5].

On 23 July 2022, the World Health Organization (WHO declared the current epidemic of MPXV disease a Public Health Emergency of International Concern (PHEIC) and called for a coordinated international response. Furthermore, as of 1 December 2022, more than 80,000 cases have been identified across more than 100 countries, with the majority in previously non-endemic countries [7,8,9]. Cases were reported outside of Africa in the United States in 2003 through imported Giant Gambian rats from Ghana that co-inhabited with a prairie dog, resulting in 53 occurrences of infection. In 2018, instances were detected in the United Kingdom and Israel through travelers coming from endemic countries. Additionally, in 2019, an MPXV case was spotted in Singapore by a man who traveled from Nigeria [3,4,5,9,10,11]. The most common modes of transmission of MPXV include direct contact with infected animal fluids (blood, urine, and saliva), skin lesions, or respiratory aerosols, and indirect transmission that could spread through contaminated inanimate objects or surfaces. Additionally, vertical transmission and fetal deaths with maculopapular lesions in the fetus have also been documented [2,3,4,9,10,11].

Recent travel to endemic regions, exposure to animals brought from endemic zones, and care providers of those infected with the MPXV have historically been related to the transmission of the disease [1,2,3,4]. However, in the 2022 outbreak, many cases have been linked to transmission between men who have sex with men (MSM) rather than a travel history to endemic areas [5,9,10,11]. Lesions and breakouts on the perianal skin were interpreted as evidence that direct physical contact with lesions during sexual contact is a probable mechanism of infection transmission. The incubation period for MPXV is generally between 6 and 13 days but can range from 5 to 21 days. Symptoms during the invasive period of the disease have occurred between 0–5 days during the ongoing epidemic with fever, backache, lymphadenopathy, and myalgia [9,10,11,12]. The eruptive stage of the disease is characterized by widespread rashes with unique morphological vesicular and ulcerative pustular lesions, mainly on the face and body, genitalia, and conjunctiva. Fatalities during the current outbreak in 2022 have remained at nearly less than 5%. However, for immunodeficient individuals, the outcomes may be poorer [4,5,9,10,11,12]. The degree to which infected individuals contributed to the MPXV outbreak is unknown. The MPXV has been found in some asymptomatic MSM cases through retrospective studies of samples collected during standard screening for sexually transmitted infections (STI), such as *Chlamydia* and *Neisseria gonorrhoeae* infections in sexual health clinics [10,11,12,13,14]. Research such as this suggests that not all MPXV patients exhibit symptoms [11,12,13]. To our knowledge, no review of relevant asymptomatic MPXV has been performed to date. Therefore, the objectives of this systematic review and meta-analysis (SRMA) are (1) to critically examine the existing literature on the asymptomatic MPXV infection cases and (2) to evaluate the global prevalence of possible asymptomatic MPXV infection cases from published studies.

## 2. Methods

### 2.1. Search Strategy and Selection Criteria

We searched six databases: PubMed, Scopus, Web of Science, ProQuest, EMBASE, EBSCOHost, and Cochrane. Preprint servers (medRxiv, arXiv, bioRxiv, BioRN, ChiRxiv, ChiRN, and SSRN) are also included in our search strategy (Appendix A). Additionally, new eligible studies were extracted by carefully searching for relevant references from included papers and other suitable reviews. Using summary estimates, the primary outcome was the percentage of asymptomatic MPXV infections among the tested and confirmed populations. The study has been registered on the International Prospective Register of Systematic Reviews (PROSPERO) as CRD42022364833.

The search keywords included “Monkeypox virus”, “MPXV”, “Monkeypox”, and “asymptomatic”. MeSH terms with an asterisk were used to identify related articles in the study title (Appendix A). The articles were saved in Mendeley Desktop V1.19.5 software to manage citations, remove duplicates, and speed up the review process. We used the term “possible asymptomatic infection”, as the definition of asymptomatic or symptomatic infection depends on the rigorous interrogation and examination of the patients regarding clinical findings over the last 3–4 weeks, as well as the laboratory (hemogram or biochemistry) assessment, which usually is not performed in someone simply denying symptoms.

### 2.2. Data Extraction and Management

Two authors individually reviewed each paper. If there was a disagreement regarding the choice of an article, two of the coauthors conversed to build consensus and agree. If there was a conflict between the two leading reviewers about the publication’s eligibility, a third coauthor was consulted to assess the article and help choose whether to include the study. The reviewers ultimately discovered four articles that were relevant to the main topic. Then from the eligible articles, the following information was gathered from each source article: the author’s name, the place where the study was conducted, the year of publication, the study design, the number of infected cases, positive cases, with a focus on capturing cases of asymptomatic MPXV. A data extraction table has been prepared in a Microsoft Excel spreadsheet for further analysis. 

Articles searched were reported using the Preferred Reporting Standard of Systematic Reviews and Meta-Analysis (PRISMA) checklist to ensure scientific precision (Figure 1). In addition, the reviewers thoroughly read all four of these publications before composing their conclusions. One of the studies by Guagliardo and colleagues [13] was divided into two articles to increase the quality of the meta-analysis since this article separately reported the prevalence of MPXV among those who had received smallpox vaccination when they were young and among participants with a history of animal exposure. Finally, five studies were included in the meta-analysis. 

### 2.3. Inclusion and Exclusion Criteria 

All articles published till October 2022 were considered for this study. As per the PICO framework, we looked for studies with adults (Population), who reportedly tested for MPXV (Intervention) and tested positive for MPXV infection (Outcome).

### 2.4. Quality Assessment

For three cross-sectional studies and one case study, two authors independently rated the studies using the Newcastle–Ottawa Quality Assessment Scale and the Joanna Briggs Institute Critical Appraisal Checklist, respectively.

### 2.5. Data Analysis

The prevalence of asymptomatic MPXV infection was calculated by dividing the number of asymptomatic cases by the total number of study participants. The heterogeneity of the studies evaluated in this meta-analysis was assessed using the I2 test. Heterogeneity is the term used to describe the degree of variance between studies. The heterogeneity was classified as low, moderate, and high, respectively, based on I2 values of less than 25%, 25–50%, and more than 50%. Therefore, the articles that were included in the meta-analysis were very heterogeneous. A random-effects model with a 95% confidence interval was used to evaluate the overall effect, and *p* < 0.05 was considered statistically significant. STATA^®^ software (version 16, STATA Corp., College Station, TX, USA) was used to conduct the meta-analysis.

## 3. Results

A rigorous search of six databases and preprint servers yielded around 178 articles, of which 25 duplicates were identified. During the preliminary screening, these duplicate papers were eliminated. The remaining 153 articles were filtered by title and abstract, and 108 were removed accordingly. Based on eligibility, 45 articles were left for the entire document screening. Twenty-nine articles were excluded because they did not meet the inclusion criteria. After all, studies were selected; 16 were chosen for qualitative synthesis, and four for quantitative synthesis (meta-analyses). For the meta-analysis, one of the studies was split into two articles based on their difference in exposure patterns. Finally, a total of five documents were quantitatively evaluated [11,12,13,14].

Table 1 provides a brief synopsis of the studies featured, and the PRISMA flow chart provides an overview of the article selection process (Figure 1). The meta-analysis contains five documents, including two cross-sectional studies, two retrospective observational studies, and one case series. Testing of cases was carried out in Belgium, France, Cameroon, and Italy. All of these investigations were carried out in countries where the spread of MPXV infections is not endemic. The most extensive study was conducted by De Baetselier and colleagues) [12] and the smallest one was by Moschese and colleagues (both in 2022) [14]. All the cases who contracted the virus were men who had sex with men except for the Cameroon study, where the disease was contracted from animals [13].

In the meta-analysis of the prevalence of possible asymptomatic cases in the studied populations, 645 individuals were evaluated, of which 66 had asymptomatic infections. The pooled percentage of asymptomatic infections in the investigated population was 9.10% (95% confidence interval [CI], 1.27–22.17%), with a high degree of heterogeneity between studies (I^2^ = 94.86%; *p* < 0.01) (Figure 2).

The findings of the quality assessment of the included studies are shown in the Appendix A. A funnel plot was not conducted due to the smaller number of studies included in the meta-analysis. However, we explicitly examine publication bias using Egger’s test for a regression intercept, which gave a *p*-value of 0.99, indicating no evidence of publication bias. Figure 3 shows the bubble plot showing the liner prediction based on the percentage of asymptomatic MPXV cases with its 95% confidence interval.

### Result Synthesis

All the cases in this systematic study were without symptoms but were shown to be positive for MPXV infection [11,12,13,14]. Two studies, one in Belgium and one in France [11,12], used polymerase chain reaction to analyze previously acquired MSM samples from standard STD testing in sexual health clinics and identified 17 asymptomatic cases of MPXV. Another cross-sectional investigation in Cameroon showed the possibility of circulating MPXV orthopoxvirus and contact with monkeypox reservoirs [13]. Some people who had not received the smallpox vaccine were IgG-positive for OPXV antibodies despite never having been afflicted with MPXV. Furthermore, people who frequented the forest were likelier to have eaten Gambian rats and were asymptomatically infected with MPXV. A case study from Milan showed two secondary asymptomatic cases with detectable MPXV in urethral discharge [14]. Real-time PCR on anal and urethral samples from confirmed MPXV cases. In this case, the reason for the testing was a close sexual encounter with a laboratory-confirmed MPXV-infected patient.

## 4. Discussion

There is no published systematic review and meta-analysis on asymptomatic MPXV infection. As mentioned earlier, the definition of an accurate asymptomatic infection in this emerging disease is complex, especially when patients present a very small number of symptoms, single skin lesions, or other subclinical findings. A total of 16 studies were reviewed for qualitative synthesis, while four studies, including 645 individuals, were selected for quantitative synthesis [11,12,13,14]. We detected that more than a tenth (10.2%), a significant proportion of those infected with the MPXV, are possibly asymptomatic cases. Because of the growing global concern, there is a resurgent concern regarding MPXV’s geographical spread and transmission mode. Efficient health response to this emergency should comprise rapid detection and appropriate management of these asymptomatic cases of MPXV infection, including more comprehensive prospective studies. Unfortunately, in the case of MPXV (unlike many other viral infections), seroprevalence studies in asymptomatic individuals are complicated as there is a lack of standardized serological tests or a shortage of tests [13,14,15].

We found that Cameroon had the highest percentage of asymptomatic infections and Belgium the lowest. Furthermore, the prevalence of asymptomatic disease in the extracted population varied between studies conducted in different geographical regions [11,12,13,14]. MPXV asymptomatic infections were unavailable in many studies because the prevalence of asymptomatic patients is unknown and is derived primarily from case series findings and retrospective investigations. The cross-sectional survey had the highest pooled percentage of asymptomatic infections, and the retrospective observational study had the lowest [11,12,13]. Our finding shows that all asymptomatic infections reported were from communities of men who engaged in sexual activity with other men.

Investigators from Belgium and France used PCR to retrospectively evaluate MSM samples acquired during standard STD testing for STDs in sexual health centers, which only may detect the virus when the infection manifest as skin lesions, where the viral loads are higher than in most clinical samples [11,12]. Furthermore, it was found that three out of four individuals who tested positive for the MPXV in the Belgian study were asymptomatic at the time of testing. Therefore, the perianal rash was misdiagnosed as a herpes outbreak. During follow-up examinations, no MPXV-infection-related clinical findings were found in the genitalia of the three people and other parts of the three people (throat, skin, among others). Interestingly, 13 asymptomatic people in the French study had positive test findings, but only two of them developed symptoms. These findings suggest that contrary to popular perception, not all MPXV patients develop symptoms [11,12,13]. Due to the recent global outbreaks, there is renewed concern about MPXV’s geographical spread and the mode of transmission of MPXV over time [1,2,3,4]. As of now, MPXV infections have spread over 10 African countries and four other continents in the last half-century [1,3,5,7]. Since 2003, MPXV infections have also expanded beyond Africa [4,5,6]. The U.S. received contaminated rats from Ghana, which had no reported human cases of the disease at the time, and this is how the disease spread throughout the country, with the U.S. having the higher number of cases [4,5,9,10].

Along with animal-to-human and human-to-animal transmission, high rates of human-to-human transmission have been reported [16]. Furthermore, the mathematical modeling of transmission from one infected human to another demonstrated the potential possibility for an epidemic of MPXV [15,16,17]. The risk of an epidemic is further escalated when there is information that the infection has been spreading undetected in non-endemic parts of the world; for example, it spread before the first instances in England were detected. Similarly, a few days before the first case was reported in Belgium, a positive sample with no signs and symptoms of MPXV infection was retrospectively obtained. It is unknown how long the virus can be detected in patient biological samples, such as the mouth, nose, anus, and genitals [14,18,19]. Our study highlights this problem that could be much wider than we estimate, requiring redoubling efforts on surveillance and research for MPXV disease.

The significance of these new case studies and retrospective studies should be considered as the asymptomatic individuals may have missed the subtle signs or may have forgotten about them, given their infection may have occurred before MPXV was on the radar [19,20]. Our finding shows that asymptomatic MPXV infections are undoubtedly prevalent; however, it is unknown how many cases lack noticeable symptoms and are therefore ignored by both patients and physicians [10,11,19,20]. Furthermore, whether patients with mild or moderate disease transmit the infection to others is not reported. Guagliardo et al. reported that asymptomatic or undiscovered MPX orthopoxvirus circulation is likely in Cameroon, and contact with MPXV reservoirs is prevalent, necessitating continued monitoring of human and animal diseases [13]. Some individuals who had not received the smallpox vaccine were found to be IgG-positive for antibodies without having previously been infected with MPXV. 

The Cameroon study researchers identified alarmingly low readings of the cycle threshold, indicating a high viral load, in patients without symptoms, but this alone does not answer the transmission question. The Belgian study by De Baetselier and colleagues reported the detection of a replication-competent virus in genital swabs of two of three asymptomatic men [11,12,13,14]. Furthermore, monkeypox-specific PCR became negative in follow-up samples obtained months later, most likely due to natural resolution. Although the three patients were not symptomatic but had a history of sexual activity at their tests, the researchers could not determine whether the virus was transmitted to their partners. In a similar study conducted in France by Ferré and colleagues [11], around 13 people reported no symptoms that could be associated with an MPXV infection, but later two people developed symptoms. More than three weeks after the initial negative anal swab, three asymptomatic participants tested positive for MPXV. Unfortunately, the researchers could not comment on the possibility of transmission because they did not try to isolate the replicating virus from the samples. Similarly, another case study from Milan [14] that used real-time PCR to test anal and urethral samples from confirmed MPXV cases found two asymptomatic secondary cases with detectable MPXV genetic material in urethral discharge, with MPXV successfully isolated from one of them. One of the two patients was completely asymptomatic, while the other reported a day of fever three days before the test date. In this case, the basis for the test was a reported close sexual encounter with a laboratory-confirmed MPXV patient and a spontaneous desire to be investigated.

In light of the unprecedented spread of the present MPXV outbreak and the atypical presentation of the associated disease, attention must be paid to the neglect of silent or mildly symptomatic viral-shedding people. If only those with clinically severe illnesses had been shedding the virus, such a widespread outbreak would not have occurred [19,20,21]. Sexual partners are unlikely to participate in high-risk intercourse with someone who is physically ill or has lesions. Therefore, information from future research that examines the clinical course of the disease through frequent, repeated medical examinations and testing among those exposed to infectious agents can help fill the critical gaps. These routine examinations may be necessary to understand these silent infections and their transmission potential [20,21]. Furthermore, it is imperative to vaccinate more individuals who are vulnerable or at higher risk and also continuously provide such groups with relevant information they may need to protect themselves and their contacts.

### Limitations

This SRMA is one of the first studies to quantify asymptomatic MPXV infections. The main strengths are its broad review of the literature on the subject and meta-analysis to establish the magnitude of the effect. The fundamental limitations of the study are the small sample size and the limited number of studies included for analysis. For example, when dealing with non-symptomatic cases during an outbreak, the risk of underestimating the non-symptomatic cases over the symptomatic ones is substantial. Our analysis may have overestimated asymptomatic cases due to the inclusion of very small studies (i.e., a case study by Moschese et al.) [14]. Above all, there is a need for an accurate and widely applicable definition of asymptomatic infection for MPXV cases globally and the need to develop highly accurate and widely available diagnostic tests to better help understand the epidemiology of the current MPXV outbreaks

## 5. Conclusions

It is unclear how common asymptomatic MPXV infections are and their role in the current outbreak. This systematic review and meta-analysis emphasize that MPVX infection can occur as asymptomatic cases, although relatively low, even considering the small number of studies available and included. These findings are preliminary due to the small number of cases in them. Asymptomatic cases may indicate viral shedding and subsequent transmission, but there is no definitive evidence and valid definition of asymptomatic MPXV infections. It is recommended that all MSM and high-risk populations be immunized to prevent the spread of MPVX infection. The tracking of contacts and the screening of high-risk populations are two other essential strategies that are needed to elucidate the proper proportion of cases of asymptomatic MPXV. 

## Figures and Tables

**Figure 1 vaccines-10-02083-f001:**
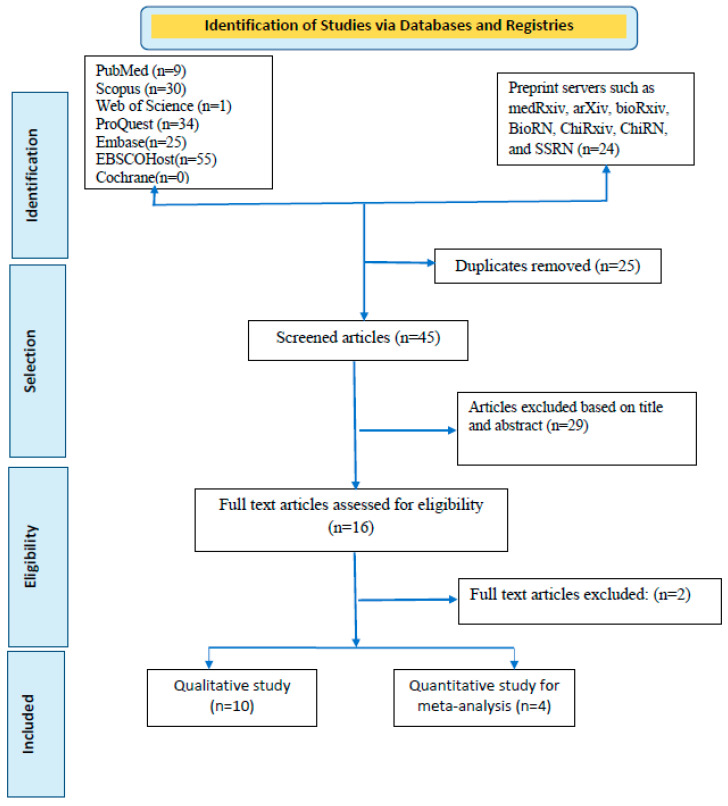
PRISMA flowchart for included studies in systematic review and meta-analysis of asymptomatic infection and transmission of monkeypox virus.

**Figure 2 vaccines-10-02083-f002:**
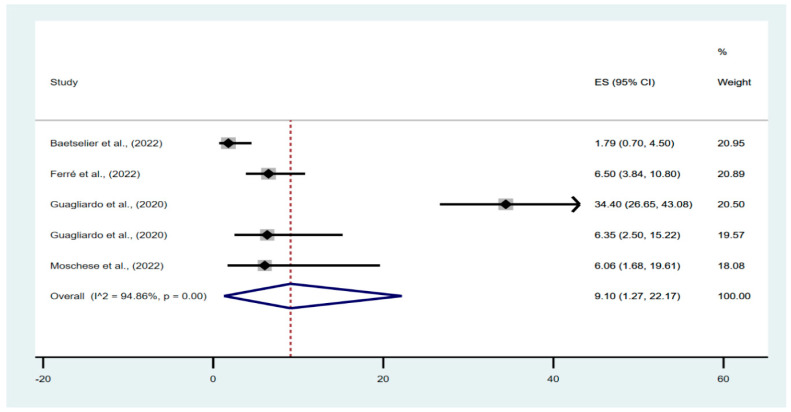
Pooled prevalence forest plot of asymptomatic monkeypox [11,12,13,14].

**Figure 3 vaccines-10-02083-f003:**
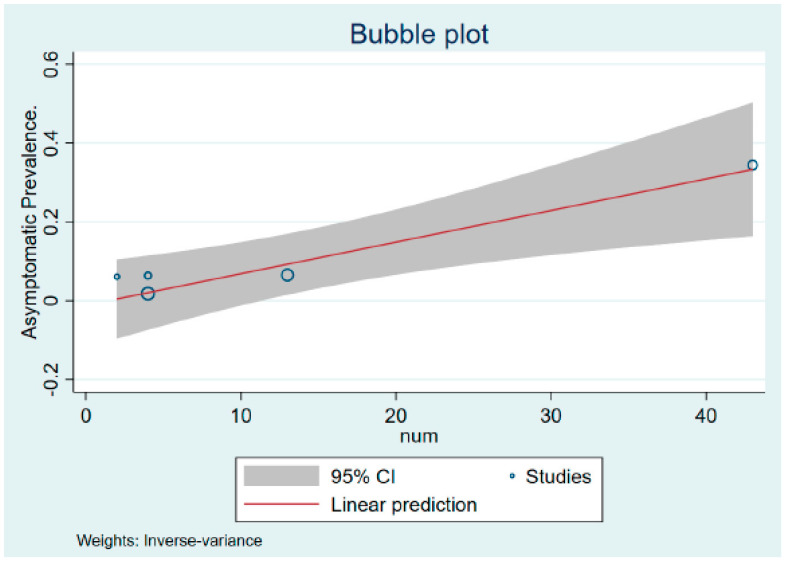
Bubble plot based on the percentage of asymptomatic monkeypox cases.

**Table 1 vaccines-10-02083-t001:** Summary of Studies Included in Meta-Analysis.

Author (Year)	Study Design	Sample	Country	Cluster	Characteristics of the Asymptomatic Cases
Baetselier et al., (2022) [12]	Retrospective observational	224	Belgium	Men who underwent sexually transmitted infection (STI) screening	1 unrecognized symptomatic case and the other 3 males were asymptomatic in the weeks before and following the sample’s collection. None of them came in touch with any known case of monkeypox, and none of their contacts experienced clinical symptoms of monkeypox. Following the initial sample, follow-up samples were collected 21 to 37 days later, by which time the monkeypox-specific PCR was negative, most likely due to the infection’s natural resolution.
Ferré et al., (2022) [11]	Retrospective observational	200	France	Asymptomatic men who have sex with men (MSM)	Out of 13, no one reported any symptoms that might be related to an MPXV infection, but 2 later came with symptoms. Three asymptomatic participants tested positive for MPXV more than 3 weeks after the initial negative anal swab.
Guagliardo et al., (2020) [13]	Cross-sectional	125	Cameroon	(1) employees of the primate sanctuary; (2) residents from four villages	There were differences in the exposure patterns to the animals that could be attributed to formal education, socioeconomic status, and the attitudes regarding conservation of animals. The findings when adjusted for the education levels of the participants showed that the individuals who visited the forest > 1 time per week were 3.4 times as likely to have eaten Gambian rats than those who visited the forest <1 visit per week. The primate sanctuary employees were less likely to have sold/touched the Gambian rats as compared to the members of the community.
Guagliardo et al., (2020) [13]	Cross-sectional	63	Cameroon	(1) employees of the primate sanctuary; (2) residents from four villages (who were not vaccinated for smallpox vaccination)	The participants were IgG positive for anti-OPXV antibodies. Among those who had not received the smallpox vaccine i.e., individuals born after 1980) had IgG positive n = 3 6.3%; IgM positive (n = 1), 1.6% were all asymptomatic for Monkeypox disease.
Moschese et al., (2022) [14]	Case study	33	Milan	Men with close sexual contact with a laboratory-confirmed monkeypox case	Except for two individuals, all patients were screened for MPXV on clinical basis suspicion. Of the two people who were not tested on a clinical basis, one was completely asymptomatic, and the other reported 1 day of fever 3 days before the testing date. In this instance, the reason for testing was a reported close sexual contact with a laboratory-confirmed monkeypox case.

## Data Availability

Documents containing all the extracted data have been made available in the manuscript and the accompanying Appendix A.

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
