# Peer review of "Potentially Asymptomatic Infection of Monkeypox Virus: A Systematic Review and Meta-Analysis"

_vaccines, 2022, doi:10.3390/vaccines10122083_

Round 1

Reviewer 1 Report

Satapathy et al reviewed "asymptomatic monkeypox" and provided a metanalysis. Generally, the review is interesting and well-written. Although this topic is not new as mentioned by authors (lines 112 and 113), I suggest publishing this interesting review because the metanalysis of asymptomatic monkeypox is not published before, to my knowledge. However, I recommend the addition of „Recommendations“ to benefit the readers. Also, I have several minor comments which are marked in the attached file. For example:

-        References are not cited according to the journal instructions.

-         Authors should revise the abbreviations throughout the manuscript,  

-        please check the journal instructions regarding the subtitles of the abstract. I think need the titles in the abstract (Background, Methods, Results, and conclusion)

-        Lines 70-21: please paraphrase.

-        Line 97: Please add this citation: Parvin, R., et al. Monkeypox virus: A comprehensive review of taxonomy, evolution, epidemiology, diagnosis, prevention, and control regiments so far. Ger. J. Microbiol. 2022. 2(2): 1-15

-        Line 98: monkeypox virus should be MPXV

-        Line 223 monkeypox virus should be MPXV

- Line 227 monkeypox virus should be MPXV

-    Line 244 monkeypox virus should be MPXV

-  Line 296:  please delete the word „monkeypox virus“ and use the abbreviation

-     Reference list, should be also revised according to the journal instructions.

Author Response

Response to Reviewers Manuscript# vaccines-2042139

We would like to thank the Editor and reviewers for the careful and thorough reading of this manuscript and for the thoughtful comments and constructive suggestions, which help to improve the quality of this manuscript. Our response follows (the reviewer’s comments are in italics).

Reviewer 1

Comment: Satapathy et al reviewed "asymptomatic monkeypox" and provided a metanalysis. Generally, the review is interesting and well-written. Although this topic is not new as mentioned by authors (lines 112 and 113), I suggest publishing this interesting review because the metanalysis of asymptomatic monkeypox is not published before, to my knowledge. However, I recommend the addition of Recommendations to benefit the readers. Also, I have several minor comments which are marked in the attached file.
Response:  We are grateful for your diligent review and guidance. We have made substantial changes to the language, grammar, and syntax. For recommendations, please see the paragraph before the limitations section.

Comment: For example:    References are not cited according to the journal instructions.
Response:  We have formatted all the references as per journal instructions and will continue to work with the proof readers if the paper is accepted. We are thankful for your insights.

Comment:       Authors should revise the abbreviations throughout the manuscript,  
Response:  We have edited and worked on abbreviations as per journal instructions and will continue to work with the proof readers if the paper is accepted. Thank you for this suggestion.

Comment:      please check the journal instructions regarding the subtitles of the abstract. I think need the titles in the abstract (Background, Methods, Results, and conclusion)
Response: We have changed our abstract subtitles to match the journal requirements.

Comment:  Lines 70-21: please paraphrase.
Response:
We have made substantial changes to the language, grammar, and syntax to rephrase this sentence and others that needed changes. Thank you.

Comment: Line 97: Please add this citation: Parvin, R., et al. Monkeypox virus: A comprehensive review of taxonomy, evolution, epidemiology, diagnosis, prevention, and control regiments so far. Ger. J. Microbiol. 2022. 2(2): 1-15
Response: we are grateful for this quality reference that you have provided and added it to our list instead of a previous one that was outdated. Thank you for your help.

Comment:  Line 98: monkeypox virus should be MPXV; -        Line 223 monkeypox virus should be MPXV; - Line 227 monkeypox virus should be MPXV; -    Line 244 monkeypox virus should be MPXV;  Line 296:  please delete the word „monkeypox virus“ and use the abbreviation
Response:  We have edited and worked on MPXV terms and abbreviations throughout the paper. Thank you for this suggestion.

Comment:    Reference list, should be also revised according to the journal instructions
Response:  We have formatted all the references as per journal instructions and will continue to work with the proof readers if the paper is accepted. We are thankful for your insights.

Reviewer 2 Report

Estimated Authors,

first of all, thank you for sharing with the present Reviewer this interesting SRMA on the occurrence of asymptomatic MPX infections.

Eventually, Authors were able to estimate a 9.1 (95%CI 1.3 to 22.2) prevalence, that is both interesting per se and useful to explain the quite rapid spreading of the MPX outbreak during 2022. 

Despite its potential interest, I've a series of concerns that have to be better addressed before the final acceptance.

1) please include a PICO item in order to make more clear the underlying question of your research, and the correspondent rationale;

2) I've some doubts about your search strategy, in this terms: we have a growing evidence that the worldwide outbreak of MPX has been associated with a substrain characterized by a very low Case Fatality Ratio (https://pubmed.ncbi.nlm.nih.gov/36355615/) as a consequence I've a little bit doubtful that including patients from African cases (i.e. the two studies from Guagliardo et al.) may be consistent with the other studies. I would suggest to perform a subanalysis by including two distinctive sub-estimates, one for non-African and the other one for African based studies.

3) when dealing with non-symptomatic cases during an outbreak, the risk for underestimating the non-symptomatic cases over the symptomatic ones is substantial, and your study design may magnificate it. In fact, you have included as source article a case study (Ref. Moschese et al.) that not only is of limited size compared to the other ones, but clearly include patients that were selected and identified in a totally different way. This flaw has to be otherwise addressed.

Author Response

Response to Reviewers Manuscript# vaccines-2042139

We would like to thank the Editor and reviewers for the careful and thorough reading of this manuscript and for the thoughtful comments and constructive suggestions, which help to improve the quality of this manuscript. Our response follows (the reviewer’s comments are in italics).

Reviewer 2

Comment:    Estimated Authors, first of all, thank you for sharing with the present Reviewer this interesting SRMA on the occurrence of asymptomatic MPX infections. Eventually, Authors were able to estimate a 9.1 (95%CI 1.3 to 22.2) prevalence, that is both interesting per se and useful to explain the quite rapid spreading of the MPX outbreak during 2022.  Despite its potential interest, I've a series of concerns that have to be better addressed before the final acceptance.
Response:   We are grateful for your encouragement, guidance, and time in reviewing this paper and helping us make it stronger.

Comment:    1) please include a PICO item in order to make more clear the underlying question of your research, and the correspondent rationale;
Response:  Thank you, we have added the PICO framework for the study.

Comment:    2) I've some doubts about your search strategy, in this terms: we have a growing evidence that the worldwide outbreak of MPX has been associated with a substrain characterized by a very low Case Fatality Ratio (https://pubmed.ncbi.nlm.nih.gov/36355615/) as a consequence I've a little bit doubtful that including patients from African cases (i.e. the two studies from Guagliardo et al.) may be consistent with the other studies. I would suggest to perform a subanalysis by including two distinctive sub-estimates, one for non-African and the other one for African based studies.
Response:  We sincerely appreciate the suggestions for a sub-group analysis (African vs non-African). Considering the small number of studies we prefer not to do a subanalysis, as there wont be much meaning of that exercise given the small and few studies. We require at least four studies in each group for a meaningful estimate. We included this as a limitation of our study.

Comment:    3) when dealing with non-symptomatic cases during an outbreak, the risk for underestimating the non-symptomatic cases over the symptomatic ones is substantial, and your study design may magnificate it. In fact, you have included as source article a case study (Ref. Moschese et al.) that not only is of limited size compared to the other ones, but clearly include patients that were selected and identified in a totally different way. This flaw has to be otherwise addressed.
Response:   We agree with you and have included your message as a major limitation. Thank you for guiding us on this problem that could arise with smaller studies and estimation. We are highly grateful to you for your insightful remarks.

Round 2

Reviewer 1 Report

Many thanks, the manuscript is improved.

Author Response

Thank you very much for your time and guidance

Reviewer 2 Report

Authors have addressed both in main text and as a rebuttal my previous concerns; therefore, I will endorse the acceptance of this study.

Author Response

Thank you very much for your time and guidance.